# Motivated reasoning in the field: polarization of prose, precedent, and policy in U.S. Circuit Courts, 1891–2013

**Wei Lu** [1]*, **Daniel L. Chen**[2]

**1** Baruch College, City University of New York, New York, New York, United States of America, **2** Toulouse School of Economics, Toulouse, Haute-Garonne, France

* wei.lu@baruch.cuny.edu

**Data availability statement:** All relevant data for this study are publicly available from the GitHub repository (https://github.com/weilu-mkt/motivated_reasoning_replication).

## Abstract

This study explores politically motivated reasoning among U.S. Circuit Court judges over the past 120 years, examining their writing style and use of previous case citations in judicial opinions. Employing natural language processing and supervised machine learning, we scrutinize how judges' language choices and legal citations reflect partisan slant. Our findings reveal a consistent, albeit modest, polarization in citation practices. More notably, there is a significant increase in polarization within the textual content of opinions, indicating a stronger presence of motivated reasoning in their prose. We also examine the impact of heightened scrutiny on judicial reasoning. On divided panels and as midterm elections draw near, judges show an increase in dissent votes while decreasing in polarization in both writing and citation practices. Furthermore, our study explores polarization dynamics among judges who are potential candidates for Supreme Court promotion. We observe that judges on the shortlist for Supreme Court vacancies demonstrate greater polarization in their selection of precedents.

"I pay very little attention to legal rules, statutes, constitutional provisions ... The first thing you do is ask yourself — forget about the law — what is a sensible resolution of this dispute? ... See if a recent Supreme Court precedent or some other legal obstacle stood in the way of ruling in favor of that sensible resolution. ... When you have a Supreme Court case or something similar, they're often extremely easy to get around." (*An Exit Interview with Richard Posner*, The New York Times, *Sep. 11, 2017*).

## Introduction

Can we quantitatively identify when judges have an easier time recruiting evidence supporting what they want to be true than evidence supporting what they want to be false [1]? This tendency is called motivated reasoning, and several recent models and experiments on

**Funding:** Work on this project was conducted while Daniel Chen received financial support from the European Research Council (Grant No. 614708) and Swiss National Science Foundation (Grant Nos. 100018-152678 and 106014-150820). The funders had no role in study design, data collection and analysis, decision to publish, or preparation of the manuscript.

**Competing interests:** The authors have declared that no competing interests exist.

motivated reasoning are summarized in [2]. Motivated reasoning is a subject of much policy debate. Does it affect real-world decision-makers? Moreover, what affects motivated reasoning? These are the core questions this paper seeks to address. Motivated reasoning is the well-documented tendency of individuals to actively seek out confirmatory information. The mechanism is said to be implicit emotion regulation, where the brain converges on judgments that maximize positive affective states associated with the attainment of motives. In controlled settings, motivation is typically inferred by the degree to which goal-related concepts are accessible in memory: the greater the motivation, the more likely individuals are to remember, notice, or recognize concepts, objects, or persons related to that goal [3] (The classic studies measure the final decision rather than reasoning [4,5]). Recently, motivated reasoning has been used to explain polarization. For example, when responding to moral dilemmas, subjects come to snap judgments and then generate a post hoc justification [6]; similarly, when interpreting data on climate change, subjects update their beliefs along party lines, particularly among those with higher cognitive reflection [7].

In prior studies of motivating reasoning in law, law student subjects are exogenously provided precedents (reasons) ([8,9]) to address the issue that differences in reasoning might be due to memory or knowledge. The experiments fix the set of precedents to choose from. Nevertheless, whether these studies on law students are externally valid to judges or other policymakers is still an open question [10]. In another study using a series of experiments on statutory interpretation with the cultural identity of parties involved as the main manipulation, [11] shows that judges and lawyers do not exhibit cultural biases, unlike law students and the general public. In our case, we show that when judges are making high-stakes decisions in Court, they could exhibit polarization in writing. Building on the framework by [12] on politically motivated reasoning, recent work by [13] provides a new design to assess politically motivated reasoning based on trust in news.

This paper explores motivated reasoning among real-world judges. Our paper sits at the intersection of constitutional law, politics, and judicial legitimacy, examining the hypothesis that judicial decision-making is politically motivated. Grounded in the debate between jurisprudential decisionism and the separation of political interests from legal procedures [14–20], we employ a quantitative approach to assess the extent to which recent shifts in the U.S. judiciary reflect broader political dynamics. Our analysis contributes to the understanding of how constitutional-legal proceduralism, often seen as a tool for upholding democratic values, may be exploited by political-economic elites to shape legal outcomes. This paper provides empirical insights into the political nature of judicial decision-making, offering a novel perspective in the context of ongoing debates on the judiciary's role in liberal democracies.

Specifically, we aim to analyze motivated reasoning using as a natural laboratory the U.S. federal courts – a high-stakes common-law space. Circuit Court judges are appointed by the U.S. president (Democrat or Republican) with life tenure. Circuit judges can introduce new legal theories (E.g., contract duty posits a general obligation to keep promises, vs. a party should be allowed to breach a contract and pay damages if it's more economically efficient than performing, also known as efficient breach theory, articulated by Richard Posner in a 1985 opinion.), shift standards or thresholds (E.g., shift from reasonable person standard to reasonable woman standard for what constitutes sexual harassment, or waive the need to prove emotional harm in court to a jury.), and rule on the constitutionality of federal and state statutes. Circuit judges provide the final decision on tens of thousands of cases per year, compared to just a hundred cases or so on the U.S. Supreme Court. Therefore Circuit decisions are the majority of what creates the law in this common-law space (and most of what law students are reading). If there is motivated reasoning among these judges, that could have

substantial legal and policy impacts. Existing research on how Democrat and Republican Circuit Court judges behave differently is extensive (see [21] for a comprehensive review), but almost all of them focus on the decisions made by judges. For example, a recent work examines how the ideology of Circuit Court judges can affect case outcomes in a wide range of Circuit cases [22]. Our paper complements this literature by shifting attention to polarization in reasoning.

Circuit courts have a handful of critical features that make them a desirable context for this empirical work. First, there is random assignment of cases to judges (who sit in panels, without juries) (This randomness has been used in a growing set of economics papers [23–28]). meaning that judges rule on similar legal issues on average. Second, there is an adversarial system where the litigants are responsible for bringing all the reasons (arguments and precedents) to a judge's attention. This means that differences in reasoning are not due to differences in knowledge (That is, we can distinguish our results from mechanical failures of inference due to bounded rationality or limited attention; in this adversarial setting, briefs bring forward all the citable reasons.). In addition, the briefs are filed prior to judicial assignment, so strategic information provision according to judge type is not feasible.

Motivated reasoning can be understood as a cognitive process where individuals distort how they interpret information to align with their pre-existing beliefs, often to reduce cognitive dissonance. [13]'s framework on politically motivated reasoning provides a clear explanation of how and why this occurs, especially in politically charged contexts. His experimental work shows that individuals do not update their beliefs in a purely Bayesian manner when receiving new information. Instead, they give greater weight to information that reinforces their prior beliefs and are more skeptical of information that challenges those beliefs. Applied to judges, this framework helps explain why judicial reasoning can exhibit ideological polarization. Even though judges are expected to be impartial, their political affiliation and identity can lead them to unconsciously distort how they interpret legal precedents or facts, much like how Thaler's subjects over-trusted news that supported their political stances. In legal contexts, where ambiguity often exists, this motivated reasoning can manifest as judges unconsciously favoring interpretations of the law that align with their political preferences. Thaler's model also predicts that the stronger an individual's prior beliefs (such as a judge's political affiliation), the more likely they are to exhibit overconfidence in the correctness of their reasoning, further reinforcing polarized judicial opinions.

We analyze over 300,000 Circuit Court opinions (representing almost a million judicial votes) from 1891 to 2013, assessing judicial reasoning through both the text of the opinions and citations to other Circuit Court rulings. These outcomes are linked to judicial biographical factors, particularly political affiliation. In this study, polarization is defined as the degree to which judicial opinions and citation patterns reflect partisan divides, with the reasoning aligning with the judge's political party. We argue that this polarization is an observable result of politically motivated reasoning, where judicial reasoning demonstrates ideological biases consistent with the judge's political preferences. Motivated reasoning often skews the decision-making process towards political beliefs, as noted by [13], leading judges to unconsciously favor legal interpretations that support their ideological views. This bias results in judges giving more weight to evidence or precedents that align with their pre-existing political beliefs, reinforcing partisan divisions in their legal reasoning. Thus, polarization in judicial opinions can be seen as a direct consequence of motivated reasoning, where cognitive biases lead judges to unconsciously align their reasoning with their political preferences.

The degree of polarization is measured by the accuracy with which we can predict a judge's political affiliation based on the content and structure of their opinions, following the approach of previous studies [29,30]. Earlier efforts to measure polarization in the text,

such as [31], might overestimate polarization in early years. Our approach improves upon this by focusing specifically on judicial texts, where political bias can be observed in both language and citation practices. Outside the legal context, [32] use the text of academic articles to predict political donations by economists. Unlike economists or members of Congress, who have discretion over the topics they address, judges are randomly assigned cases. This ex-ante assignment makes judicial reasoning a more direct reflection of political bias. Moreover, political donations are made after (ex-post) economist writings, while the political affiliation of the appointing party is determined before (ex-ante) judicial writings. We seek to predict a predetermined measure of ideology prior to reasoning on the case.

Compared to the extant literature on political polarization, the contribution of our paper is twofold. Firstly, we study how polarization is manifested in both judicial reasoning and policy decisions. While research on the latter is abundant [33], little is known about how the reasoning process of judges can inform their political preferences. By examining the language and citation patterns in judicial opinions, we provide new insights into how judges' political affiliations may influence their legal reasoning, complementing the existing literature that primarily focuses on judicial votes and outcomes.

Secondly, we leverage recent advances in machine learning and natural language processing to measure motivated reasoning using a very large dataset of citations and texts. Compared to previous studies that rely on votes or appointment variables, the adoption of these methods allows us to make use of the high-dimensional dataset on judicial behavior to answer our question of interest. This approach is particularly valuable given the complex nature of legal texts [21], as it enables us to uncover nuanced patterns of polarization that may not be readily apparent using traditional methods.

In less technical terms, we "train" the machine-learning model on a large set of texts where we already know the judge's political affiliation. The model then learns which linguistic features (words, phrases, types of arguments, etc.) are more frequently used by judges from each political party. Once trained, the model can take a new judicial opinion, analyze its language, and estimate how likely the judge is to align with a particular political ideology based on the patterns in their writing.

This measure of polarization reflects how much the language in an opinion resembles the typical language used by judges of a specific political party. If the opinion strongly mirrors the patterns associated with one party, we interpret this as evidence of politically motivated reasoning. Conversely, if the opinion uses language that doesn't clearly align with any political party, it would suggest a lower degree of polarization.

A new contribution is to look at the polarization of precedent, as these are the legal reasons cited to justify a decision. We use a network of citations to previous Circuit Court decisions to predict partisan affiliation. Unlike the case of prose, we find low yet steady levels of precedent polarization over time, indicating that judges tend to express ideological differences through writing instead of choices of precedents in our context. These results complement previous work with smaller samples by [34] showing that circuit judges tend to cite judges from the same party, and that of [35] showing that circuit judges tend to cite Supreme Court cases authored by judges from the same party.

Finally, we look at how the polarization in prose and precedent changes when judges are under more scrutiny. Specifically, we examine two such scenarios: The first is whether a judge sits on a divided panel of judges from both parties. The second scenario is whether the opinions were filed when the midterm or presidential elections were close. Some research suggests that the threat of actual "whistleblowing" tempers the decisions issued when under scrutiny [36,37], as reflected by the increase in dissents. Consistent with this interpretation,

the polarization in text and citations reduces when under scrutiny. Moreover, we examine how polarization varies when judges have promotion incentives. We find that judges exhibit more polarization in precedent when they are a contender for a Supreme Court vacancy.

## Measuring motivated reasoning in judicial context

In this paper, we define "motivated reasoning" in the judicial context as the ability to predict a judge's political affiliations based on the way they write opinions and cite precedents. Existing literature in economics has shown that the predictability/the prediction accuracy of texts and other behaviors can be used as valid measures of party differences and cultural distance [29,30]. Intuitively, if we can infer a judge's political affiliation from his or her reasoning process, it is probable that political considerations is playing a role in such process, especially in precedents where judges have the discretion to select the relevant cases.

[12] provides a theoretical framework for this measure. They propose that politically motivated reasoning in our setting can be defined as the distortion of how political dispositions (political party) affect the way a judge interprets new evidence (cases) to update his prior beliefs to form a posterior (reflected by texts and citations in opinions). Three features of the institutional setting ensure that the predictability measure we have is not related to varying priors or new evidence, and the predictability of political parties can be attributed to the distortion caused by political dispositions.

Firstly, the style and content of a judge's opinions, along with their chosen citations, offer insights into their formal reasoning processes. Prior literature has shown that judicial fact discretion, how judges believe and interpret the facts presented, can be related to the identity of judges [38]. Since how judges recruit precedents and prose in their opinions constitutes the judicial opinion, we would be observing any slant in the formal reasoning process made explicit in their opinion.

Secondly, the cases are assigned quasi-randomly to judges (Some research suggests that a few of the courts do not assign cases to judges completely randomly, but the reasons for non-random assignment include workload, scheduling, and professional development[39]. There is no direct evidence that political party is related to the assignment of cases.). The as-if random assignment of cases means that every judge will on average see a similar variety of cases. Notably, cases that should cite certain precedents or refer to certain topics should not systematically differ across judges due to this random assignment process.

Thirdly, absent politically motivated reasoning, the reasoning in the cases should be non-partisan because judges are asked that they "*not be swayed by partisan interests*"(http://www. uscourts.gov/judges-judgeships/code-conduct-united-states-judges). If judges follow this edict, then a reasonable guess on party affiliation based on the opinion is 0.5 – i.e., the probabilities of the writer being a member of Republican Party or a Democratic Party should be the same. However, this might diverge if judges are systematically interpreting the facts and the law in a different manner that is reflective of their political party.

If the expressed reasoning of judges is motivated by partisan views, then the choice of language and citations might be informative of the political party. Motivated reasoning can alter the way judges interpret and evaluate information from briefs and precedents, which would lead to differences in their expressed arguments. If a judge's political affiliations can be predicted based on their writing and citations to legal precedent, it would suggest that their reasoning can be influenced by their political leanings.

## Data

Our dataset includes a collection of 318,474 opinions published by U.S. Circuit Courts from 1891 to 2013 based on [37]. We limit our analysis to opinions written by one judge, excluding opinions labeled *per curiam*, which are authored by the whole panel without designating a specific author, and opinions drafted by multiple judges. Among all opinions, 279,167 are majority opinions and 26,441 are dissent opinions. For each opinion, we observe the full text, legal precedents, as well as all votes cast by judges on the panels for each case. We focus on precedents of previous Circuit Court opinions and the partisan policy is constructed using data on judge dissenting votes.

To study the heterogeneity of motivated reasoning across judicial characteristics, we link the opinions to the United States Courts of Appeals Databases and Attributes of the United States Federal Court Judges from [40], and use variables such as political affiliations of judges, Circuit Court, the political composition of panels, year, quarter to presidential elections for subsequent analysis.

## Classification

Since the predictability of judicial reasoning serves as our measure of motivated reasoning, we conceptualize this measurement problem as equivalent to a binary classification problem in machine learning using high-dimensional text and citation data as inputs and the political affiliations of judges as the outcome variable. In our two prediction tasks, we aim to predict the political affiliation of circuit court judges, specifically whether they belong to the Democratic or Republican party. The affiliation is represented by a binary variable: "1" for Democratic judges and "0" for Republican judges. We use opinion texts and citation embeddings as our predictors. To determine a judge's average stance over a year, we average these embeddings. Our goal is to predict the likelihood that a judge's political affiliation matches their true party affiliation.

We use sample splitting to avoid overfitting the models. In Text Classification, we randomly chose 10% of our dataset, which is about 31,000 opinions as our sample dataset due to computational constraints. Afterward, 30% of the 10% sample is used as the test set and the remaining 70% as the training set. For Citation Classification, we use the full dataset as our sample dataset, with 30% as the test set. The test set is only used after training to assess the performance of the algorithms. The best model (an ensemble of models with best-performing parameters) in each task will be applied to the full sample to generate predictions for all opinions. The remaining strategies and training details are in the S1 Text.

## Training algorithms for texts

Using texts for predictions is not a trivial task, especially given the amount of words and documents in our sample. We leverage recent advances in natural language processing to classify political affiliations of Circuit Court judges by fine-tuning pre-trained large language models using opinion texts directly as inputs. In recent years, transformer-based pre-trained large language models have been proven to have satisfactory performance on a variety of NLP tasks. Even with a small sample for fine-tuning, pre-trained models can further significantly improve the performance [41]. In this paper, we will use an ensemble of several commonly-used pre-trained transformer models to ensure the robustness of our results by averaging the predictions across models. Before fine-tuning, for each opinion, we use the Microsoft Presidio

tool [42] to detect and replace all names (including judges and any person's names) and locations to the word "PERSON" and "LOCATION". Doing so will prevent the pre-trained models from relying on the name and location information for classification, a problem known as data leakage. After that, the first 512 words of each opinion, which is the maximum length allowed by models, will be used as inputs for pre-trained models to learn.

## Training algorithms for citations

For precedents, we combine network representation models with ensemble supervised learning for classification. In the first step, we construct and transform the citation network into dense low-dimensional vectors. Specifically, we create a weighted directed graph of 310,282 nodes, and transform the citation network into citation embeddings of 300 dimensions using the node2vec algorithm by [43]. The node2vec algorithm rests on the idea that a word is represented by its "neighboring words" [44] in natural language processing. It adopts a random walk approach across the network to generate sequences of citations, and by maximizing the probability of neighboring citations, we can have latent vectors that "*maximize the likelihood of preserving network neighborhoods*" of citations. Then, we use an ensemble of commonly used supervised machine learning algorithms as our prediction algorithm, consistent with similar strategies used in previous literature [45].

## Permutation inference

To ensure that the algorithms are indeed learning from the training set, we generate a random permutation of political parties with an equal probability of two parties for all authors and use this list as the dependent variable for training. If the algorithms are learning correctly from the data, using random series as the dependent variable should result in random predictions. A similar strategy is also implemented by [29], who randomly shuffle the share of Republicans/Democrats in Congress during the year in which a particular congressman. In practice, we train another set of models with the same parameters on the random series for both Text and Citation Classification.

## Results

### Polarization in prose and precedents across time

We begin with an overview of how polarization in prose and precedent evolves over time. Fig 1 illustrates the trend in average polarization levels within opinion texts. The magnitude of average polarization in writing consistently exceeds 0.5 and surged towards 0.8 after 1950. These trends imply an increasing propensity for motivated reasoning among judges when drafting their opinions. Although [29] identified an increase in polarization in congressional speeches after 2000, it is significantly lower compared to the polarization observed in judicial opinions. To put this effect size in perspective, in [29], the polarization varies between 0.5 and 0.515. Notably, the placebo test involving random shuffling series aligns closely with the 0.5 benchmark, validating our models' ability to produce random predictions when analyzing data with randomly permuted party affiliations of judges. The fact that polarization was more present a century ago is consistent with other analyses of partisan behavior in the judiciary [46]. In the analyses that follow, we demonstrate how scrutiny influences this measure of partisanship, taking into account the specific time period in question.

Furthermore, we investigate the presence of motivated reasoning in the selection of precedents by judges. Fig 2 shows that, over the past 120 years, Circuit Court judges have consistently demonstrated a lower level of motivated reasoning in their choice of legal precedents,

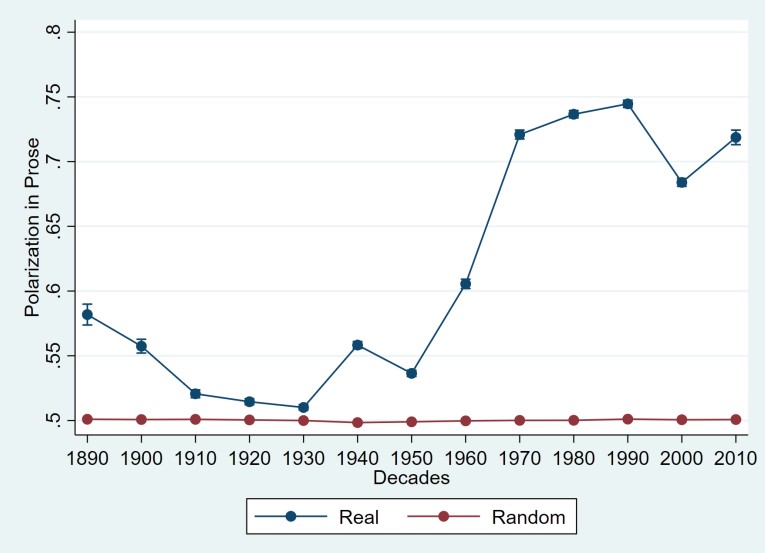

**Fig 1. Polarization in prose in U.S. Circuit Courts.** *Notes:* Polarization measures over time in U.S. Circuit Courts, 1891–2013 for writing. The blue line gives the average polarization in the true dataset. The red line gives the average polarization in the shuffled dataset (random party affiliations). Error bars indicate the 99% confidence interval.

especially when compared to the more pronounced motivated reasoning observed in texts. This suggests that, unlike the choice of language, judges are more constrained in their choice of precedents. However, the level of polarization is still distinguishable from the placebo random series of 0.5 and higher than the polarization in congressional speech found in [29].

The higher levels of polarization in prose compared to precedents indicate that judges may have more discretion in how they frame and articulate their arguments than in their selection of legal authorities. One potential explanation might be the rhetorical style of judicial over-stating, a product of cognitive processes and a means to enhance the judiciary's legitimacy [47]. On the other hand, Circuit Court judges may be constrained by precedents as they face the reversal from higher courts if deviating too much from precedent [33].

## Polarization in reasoning or decision when judges are under greater scrutiny

In this section, we examine how judicial reasoning in prose and precedents evolve under scenarios of increased judicial scrutiny. We hypothesize that when judges face such scrutiny—particularly in politically diverse settings—they may suppress overt expressions of motivated reasoning in their written opinions, even if their final decisions (as reflected in voting behavior) remain polarized. To examine this, we analyze both judicial prose and precedent citations, as well as voting patterns, which have consistently demonstrated partisan tendencies.

**Divided panels.** Our investigation begins with how judicial reasoning and decisions are influenced when judges serve on a three-member panel that consists of both Republican and Democratic appointees. According to [13]'s framework, motivated reasoning occurs when individuals process information in a way that aligns with their prior beliefs, especially in politically charged environments. In judicial settings, motivated reasoning would predict that judges are likely to favor legal interpretations that support their political ideology. However,

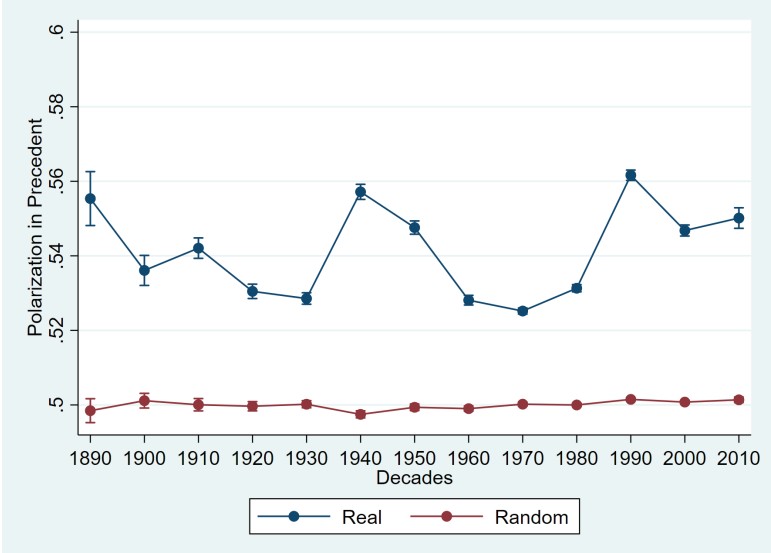

**Fig 2. Polarization in precedents in U.S. Circuit Courts.** *Notes:* Polarization measures over time in U.S. Circuit Courts, 1891-2013 for citations. The blue line gives the average polarization in the true dataset. The red line gives the average polarization in the shuffled dataset (random party affiliations). Error bars indicate the 99% confidence interval.

when judges are part of politically divided panels, they face heightened scrutiny from colleagues with opposing ideological views. This scrutiny leads to two distinct but connected effects: it moderates how judges express their reasoning in written opinions, but it does not eliminate ideological disagreements, which manifest as dissenting votes.

Judges under scrutiny may temper their written opinions to avoid overt expressions of partisanship that could be easily criticized by opposing panel members. This occurs because the diverse panel composition forces judges to justify their legal reasoning in a way that can withstand scrutiny from colleagues with different ideological viewpoints. As a result, we observe a reduction in polarization in the language and legal citations of their opinions.

However, while motivated reasoning may be constrained in the written text, ideological differences still exist at a deeper level. These differences are harder to suppress in the final decision-making process, which often results in dissenting votes. Dissent, in this case, becomes a way for judges to express their underlying political beliefs when they cannot fully align with the majority opinion. Voting behavior, unlike written opinions, is less subject to the same degree of scrutiny because it is a direct reflection of each judge's individual stance on the case outcome. Therefore, even though judges moderate their reasoning to align with doctrinal standards under scrutiny, they are still likely to cast dissenting votes when their core ideological beliefs diverge from the majority. Moreover, previous studies indicate that political divisions within a panel creates an opportunity for whistleblowing, through dissenting opinions, to expose disobedient decision-making by the majority. In the presence of such a whistleblower, the majority must sometimes capitulate and keep its decision within the confines of doctrine [22,28,36].

To explore these dynamics, we employ a linear regression model that examines polarization in judicial reasoning and dissenting votes, with the key independent variable being participation in a politically divided panel. The counterfactual group consists of judges participating in politically homogeneous panels. This model controls for Circuit × Year and legal issues

fixed effects. In Table 2, we also look at the scenario of whether a judge sitting on a divided panel is a minority judge in terms of political affiliations. In this situation, heightened scrutiny will lead to even greater moderation in reasoning, considering the composition of the panel.

Tables 1 and 2 show that judges are more likely to cast dissenting votes when they are part of a divided panel or as a minority member within such a panel. Concurrently, our findings indicate a reduced degree of motivated reasoning in their prose and precedent citations, aligning with our hypothesis. Compared to the previous literature on congressional speech [29] where the polarization variation is only 0.015, the dampening effect we observe is large.

**Electoral cycles.** We observe a similar pattern in another scenario of scrutiny, namely, the periods leading up to Presidential and midterm elections. With heightened scrutiny, we may expect a decrease in polarization with their reasoning. To investigate this hypothesis, we divided the time into 16 quarters preceding a Presidential election. Our analysis focuses on the electoral cycles and their impact on polarization in judicial reasoning. The results of this analysis are detailed in Table 3.

We find that judges reduce polarization in both the texts of opinions and citations preceding midterm elections, indicating a notable shift in judicial behavior during these periods. This trend contrasts with the pattern observed before Presidential elections, where such polarization does not exhibit a significant change. A plausible explanation for this discrepancy can be linked to the heightened partisan political priming associated with Presidential elections. According to [46], this kind of priming is intense during Presidential elections, potentially neutralizing the judges' inclination to reduce polarization under scrutiny.

**Table 1. Polarization in divided panels.**

|  | (1) | (2) | (3) |
|---|---|---|---|
|  | Text | Citation | Dissent vote |
| Divided Panel | −0.032*** | −0.038*** | 0.006*** |
|  | (0.005) | (0.004) | (0.001) |
| Observations | 310604 | 269155 | 1030343 |
| $R^2$ | 0.335 | 0.125 | 0.009 |
| Circuit × Year FE | ✓ | ✓ | ✓ |
| Legal Issue FE | ✓ | ✓ | ✓ |

*Notes:* This table shows how judges on a divided panel would exhibit polarization in prose, precedent, and policy. The unit of observation for Column (1) and (2) is at the opinion level, and Column (3) is at the vote level. Every case has three votes from three judges sitting in a panel and judges are allowed to write concurring or dissent opinions besides the majority opinion for each case. We controlled for Circuit × Year and legal issues fixed effects. Standard errors clustered at judge level in parentheses.* $p < 0.1,$** $p < 0.05,$*** $p < 0.01$

**Table 2. Polarization in divided panels.**

|  | Text | Citation | Dissent Vote |
|---|---|---|---|
|  | (1) | (2) | (3) |
| Minority | −0.020*** | −0.030*** | 0.012*** |
|  | (0.006) | (0.004) | (0.001) |
| Observations | 225817 | 196097 | 742495 |
| $R^2$ | 0.320 | 0.065 | 0.012 |
| Circuit × Year FE | ✓ | ✓ | ✓ |
| Legal Issue FE | ✓ | ✓ | ✓ |

*Notes:* Effect of being a minority judge (D of DRR or R of RDD) on the polarization in texts and citations, and the likelihood to cast a dissenting vote, controlling for Circuit × Year and legal issues fixed effects. The unit of observation for Column (1) and (2) is at the opinion level, and Column (3) is at the vote level. Standard errors clustered at judge level in parentheses. The sample is cases with judges from both political parties. $*p < .1, * * p < 0.05, * * *p < .01.$

**Table 3. Electoral cycles in text and citation.**

| | (1) | (2) | (3) | (4) | (5) |
|---|---|---|---|---|---|
| | Prose | | Precedents | | Dissent Vote |
| | All Op | Dis Op | All Op | Dis Op | |
| Quarter to election = 1 | 0.004 | −0.019 | −0.002 | 0.007 | 0.005** |
| | (0.005) | (0.013) | (0.003) | (0.006) | (0.002) |
| Quarter to election = 2 | 0.005 | −0.021 | 0.003 | 0.001 | 0.003* |
| | (0.004) | (0.013) | (0.002) | (0.006) | (0.002) |
| Quarter to election = 3 | 0.009** | −0.003 | 0.004** | 0.002 | 0.003* |
| | (0.004) | (0.011) | (0.002) | (0.006) | (0.002) |
| Quarter to election = 4 | 0.010** | −0.019 | 0.001 | −0.014** | −0.001 |
| | (0.004) | (0.013) | (0.002) | (0.006) | (0.002) |
| Quarter to election = 5 | 0.010* | 0.004 | −0.001 | −0.010 | 0.002 |
| | (0.006) | (0.017) | (0.003) | (0.008) | (0.002) |
| Quarter to election = 6 | 0.005 | −0.006 | 0.000 | −0.011 | 0.001 |
| | (0.006) | (0.017) | (0.003) | (0.008) | (0.002) |
| Quarter to election = 7 | 0.006 | 0.008 | −0.001 | −0.008 | −0.001 |
| | (0.006) | (0.017) | (0.002) | (0.008) | (0.002) |
| Quarter to election = 8 | −0.008* | −0.025 | −0.002 | −0.019** | 0.001 |
| | (0.005) | (0.015) | (0.002) | (0.007) | (0.002) |
| Quarter to election = 9 | −0.012** | 0.001 | −0.002 | −0.018** | 0.005* |
| | (0.006) | (0.019) | (0.003) | (0.009) | (0.002) |
| Quarter to election = 10 | −0.012** | −0.009 | −0.001 | −0.021** | 0.003 |
| | (0.006) | (0.019) | (0.003) | (0.009) | (0.002) |
| Quarter to election = 11 | −0.006 | −0.010 | 0.000 | −0.016* | 0.003 |
| | (0.005) | (0.019) | (0.003) | (0.009) | (0.002) |
| Quarter to election = 12 | −0.011*** | −0.015 | −0.003 | −0.010 | −0.001 |
| | (0.004) | (0.013) | (0.002) | (0.006) | (0.002) |
| Quarter to election = 13 | −0.003 | −0.021 | 0.000 | −0.004 | 0.000 |
| | (0.005) | (0.015) | (0.002) | (0.007) | (0.002) |
| Quarter to election = 14 | −0.009* | −0.031** | 0.000 | −0.000 | −0.002 |
| | (0.005) | (0.015) | (0.002) | (0.007) | (0.002) |
| Quarter to election = 15 | −0.002 | −0.021 | 0.002 | 0.003 | −0.002 |
| | (0.004) | (0.013) | (0.002) | (0.006) | (0.002) |
| Observations | 190135 | 17110 | 178609 | 13494 | 606999 |
| $R^2$ | 0.243 | 0.137 | 0.097 | 0.086 | 0.008 |
| Circuit × Year FE | ✓ | ✓ | ✓ | ✓ | ✓ |
| Season FE | ✓ | ✓ | ✓ | ✓ | ✓ |
| Legal Issue FE | ✓ | ✓ | ✓ | ✓ | ✓ |

*Notes:* The unit of observation for Column (1) to (4) is at the opinion level, and Column (5) is at the vote level. Standard errors clustered at judge level in parentheses. The base period is 16 quarters to Presidential Elections. The sample is cases published after 1975. $^*$ $p < 0.1$, $^{**}$ $p < 0.05$, $^{***}$ $p < 0.01$

Although U.S. Circuit Court judges enjoy life tenure, they do not operate in a vacuum. The argument for reduced polarization near elections hinges on the heightened scrutiny that judges face during these periods. Key stakeholders—such as political actors, legal professionals, and the media—pay particularly close attention to judicial decisions with potential political ramifications, creating indirect pressure on judges to appear impartial or less overtly partisan [34]. This scrutiny may heighten judges' awareness of how their rulings align with party politics, prompting them to strategically temper their written reasoning to avoid accusations of partisanship or backlash—especially in high-stakes election contexts [33,36]. Even the presence of a single judge from the opposing party on a panel can lead to "whistleblowing" through dissent, raising the reputational stakes in a close-knit appellate community. Furthermore, although reversal by the Supreme Court remains statistically rare, judges seek to avoid

decisions that might appear excessively outcome-driven and invite negative attention [33]. Career advancement incentives add another layer: [48] find that judges who appear on shortlists for promotion sometimes alter their judicial behavior, a pattern that parallels evidence of electoral-cycle adjustments in other judicial domains [37,46]. More broadly, judges' concerns about professional standing, institutional legitimacy, and adherence to norms of impartiality [12,22,28] can all contribute to a moderation in overt partisanship near elections, helping explain the decline in polarization we observe in both opinion text and citations leading up to midterm contests.

Furthermore, an intriguing pattern emerges in the context of both midterm and Presidential elections as they draw near: an increase in dissenting votes. This observation, documented in Column 5, aligns with what can be described as a 'whistleblowing effect' similar to that found on politically divided panels. During periods of increased scrutiny, which are common around election times, judges might express dissent more openly, a behavior that is consistent with a whistleblowing response.

To summarize, we investigate two different situations of increased scrutiny, which prior research has suggested can lead to greater dissent in votes. While our analysis confirmed this finding, we observe a reduction in polarization in prose and citations to precedent at the same time. Such a response suggests a complex interplay between the political environment and judicial decision-making, that judges will tend to exhibit partisanship in decisions rather than in reasoning.

To address concerns around non-random assignment, we have performed robustness checks by excluding the Second, Eighth, Ninth, and D.C. Circuits, which previous research identifies as potentially less random in panel assignments [49]. Our findings remain consistent even after these exclusions, shown in S1 Text S4 and S5 Tables. Furthermore, [50] provides evidence suggesting that panel compositions do not exhibit time-based autocorrelation, further reinforcing the validity of this assumption. We are also aware that within-panel dynamics could influence opinion content. However, we conduct robustness checks focused on senior judges who oversee opinion assignments to account for potential multi-judge influence, shown in S6 and S7 Tables (S1 Text). Our results remain consistent, supporting our assumption that we can attribute opinion text predominantly to a single judge.

## Polarization and promotion incentives

In this section, we analyze an institutional factor that is likely to influence political polarization in the courts: promotion incentives. We concentrate on the nomination process for Supreme Court of the United States (SCOTUS) justices, where Circuit Court judges are potential candidates for elevation to the highest court by presidential and senate appointment. This scenario raises a question: do judges demonstrate increased partisan polarization in their reasoning and decision-making as a strategy to secure a nomination? Drawing on the findings of [48], who observed that judges on the president's "shortlist" are more likely to write dissent opinions and vote in line with the presidents, our analysis seeks to understand if politically motivated reasoning might change with SCOTUS vacancies using a much larger sample of judges and years. Detailed methodology and data processing information for this analysis are provided in the S1 Text.

In Table 4, we present our results using the same specification as in Table 1. The results indicates no systematic differences between judges on the presidential shortlist and their non-contender counterparts; moreover, a Supreme Court vacancy does not result in significant changes in behavior across the entire judicial spectrum in Circuit Courts. However, during a Supreme Court vacancy, contender judges demonstrate noticeably more polarization in their

**Table 4. Polarization in SCOTUS vacancies.**

|  | Text | Citation | Dissent Vote |
|---|---|---|---|
|  | (1) | (2) | (3) |
| Vacancy | −0.001 | −0.001 | 0.000 |
|  | (0.003) | (0.001) | (0.001) |
| Contenders | −0.041 | −0.005 | 0.003 |
|  | (0.039) | (0.021) | (0.005) |
| Vacancy × Contenders | 0.016 | 0.017** | −0.001 |
|  | (0.019) | (0.008) | (0.006) |
| Observations | 49,711 | 46,759 | 153,672 |
| $R^2$ | 0.257 | 0.100 | 0.008 |
| Circuit × Year FE | ✓ | ✓ | ✓ |
| Legal Issue FE | ✓ | ✓ | ✓ |

Standard errors in parentheses

* $p < 0.1$,** $p < 0.05$,*** $p < 0.01$

*Notes:* Effect of being a SCOTUS vacancy contender on the polarization in texts and citations, and the likelihood to cast a dissenting vote, controlling for Circuit × Year and legal issues fixed effects. The unit of observation for Column (1) and (2) is at opinion level, and Column (3) at the vote level. Standard errors clustered at judge level in parentheses. Sample is cases with judges from both political parties after 1975. * $p < .1$, ** $p < 0.05$, *** $p < .01$.

selection of legal precedents. Nevertheless, we observe no significant extension of this trend to their writing style or voting patterns, which differs from [48], whom observed a significant effect for dissent votes for contenders. From a theoretical standpoint, it is remarkable that contender judges choose to standout to a potential nominating president through their citations to precedent, which we previously documented to be less polarizing in general. Presidents may look to nominate partisan/ideological allies rather than individuals that are politically ambiguous or moderate in their behavior in how they follow precedents. The finding here suggests that the reasoning process of judges, just like decisions, might also be strategic, depending on interests and scrutiny involved.

## Conclusion

Judges are nominally expected to sit above the partisan fray. However, we find they are divisive in their rhetoric and citations to legal precedent. We find that both text and citations display polarization, with text being even more polarized. In addition, judges display less polarization in reasoning when under greater scrutiny, sitting on divided panels, or before elections. Collectively, these findings suggest a divergence in how judges approach their reasoning and decision-making processes, reflecting varying degrees of partisanship under different circumstances.

Lifetime-appointed judges assert that their decisions are not influenced by politics. However, their voting trends and the intense partisan struggles during confirmation processes suggest otherwise. Our findings reveal the political nature of judicial reasoning measured in their rhetoric and their citations to precedent. If judges cherry-pick their precedents, this casts a shadow over the fairness of their decisions. A diminished sense of legitimacy can lead to decreased compliance with the law, which can have social and economic implications. Trust has been shown to have impacts, see [51]'s recent paper documenting this link causally. They show that enhanced trust spurs reliance on formal institutions. Reliance on formal institutions can, in turn, propel economic development, investments, and entrepreneurial undertakings. While our paper may not directly quantify these effects, it seeks to underscore their significance.

## Supporting information

**S1 Text.**
(PDF)

## Author contributions

**Conceptualization:** Wei Lu, Daniel L. Chen.

**Data curation:** Daniel L. Chen.

**Formal analysis:** Wei Lu, Daniel L. Chen.

**Investigation:** Wei Lu, Daniel L. Chen.

**Methodology:** Wei Lu, Daniel L. Chen.

**Writing – original draft:** Wei Lu, Daniel L. Chen.

**Writing – review & editing:** Wei Lu, Daniel L. Chen.

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
