## [Decision Letter · Decision Letter 0]

23 Sep 2024

PONE-D-24-34043Motivated Reasoning in the Field: Polarization of Prose, Precedent, and Policy in U.S. Circuit Courts, 1891-2013PLOS ONE

Dear Dr. Lu,

Thank you for submitting your manuscript to PLOS ONE. After careful consideration, we feel that it has merit but does not fully meet PLOS ONE’s publication criteria as it currently stands. Therefore, we invite you to submit a revised version of the manuscript that addresses the points raised during the review process.

We have received reports from two reviewers who are experts in the field. While they appreciate the great potential of your analysis, both point towards shortcomings and need for clarification in the current manuscript. Especially, some theoretical and conceptual elements of the study are criticized as being underdeveloped. I would ask the authors to pay particular attention to these issues.

We look forward to receiving your revised manuscript.

Kind regards,

Jerg Gutmann

Academic Editor

PLOS ONE

Journal Requirements:

“Work on this project was conducted while Daniel Chen received financial

support from the European Research Council (Grant No. 614708) and Swiss National Science Foundation (Grant Nos. 100018-152678 and 106014-150820).”

“Work on this project was conducted while Daniel Chen received financial support from the European Research Council (Grant No. 614708) and Swiss National Science Foundation (Grant Nos. 100018-152678 and 106014-150820).

“Work on this project was conducted while Daniel Chen received financial

support from the European Research Council (Grant No. 614708) and Swiss National Science Foundation (Grant Nos. 100018-152678 and 106014-150820).”

“None”

Reviewers' comments:

Reviewer's Responses to Questions

**Comments to the Author**

1. Is the manuscript technically sound, and do the data support the conclusions?

Reviewer #1: Yes

Reviewer #2: Partly

2. Has the statistical analysis been performed appropriately and rigorously? 

Reviewer #1: Yes

Reviewer #2: Yes

3. Have the authors made all data underlying the findings in their manuscript fully available?

Reviewer #1: Yes

Reviewer #2: Yes

4. Is the manuscript presented in an intelligible fashion and written in standard English?

Reviewer #1: Yes

Reviewer #2: Yes

5. Review Comments to the Author

Reviewer #1: Motivated Reasoning in the Field: Polarization of Prose, Precedent, and Policy in U.S. Circuit Courts, 1891-2013

This paper presents original results of a rigorous analysis with conclusions that are supported by the data. The authors are correct that much of the literature examines solely dispositional outcomes and votes. I outline some questions/issues that the authors might consider addressing in their revisions:

-- The paper may benefit from a clearer explication of the concept of polarization in this context. Scholars often conceive of it as a reflection of disparate (or diverging) policy preferences. Thus, as a conceptual matter, are the authors using the opinion text as an indicator of polarization, as rooted in judges’ preferences? Alternatively, is it more appropriate to conceive of the text-based indicator of polarization as merely the reflection of judges’ behavior in each case, which may or may not reflect their sincere preferences?

-- Although the whistleblowing argument (for the analysis of politically divided panels) makes some sense, the argument for expecting less polarization as elections near is not particularly clear. I am not sure why unelected judges would necessarily be all that concerned about the appearance of their opinions right before an election. The public pays little (and often no) attention to lower court judges, absent a few highly publicized cases. So, why should an approaching election make them more sensitive? It seems like any concern for the appearance of an opinion would depend only on the case context.

-- The authors make several potentially problematic assumptions that they might want to further defend/explore. First, they emphasize the randomness of panel selection in the U.S. Courts of Appeals as an essential foundation for their research design. However, some research has questioned whether panel randomness exists in practice – see, e.g., Levy & Chilton 2015 and Levy 2020 (Cornell Law Review). And, it may be that non-randomness is more prevalent in certain circuits (e.g., the Ninth Circuit). The authors briefly note this possibility in footnote #6, but they ought to address it earlier (e.g., they feature random panel selection at the beginning of the paper, only to hedge and call it “quasi-random” several pages later). Also, while the authors argue that political party appears unrelated to panel assignment, it could be that certain judges exhibit nonrandom selection based on the policy issue, which might correlate with partisanship. Thus, I think the authors should do more to scrutinize the assumption that any nonrandom selection is unrelated to party.

Next, the authors focus on opinions “written by one” judge while dropping per curiam opinions and those they claim to identify as “drafted by multiple judges.” The author’s key assumption is that they can attribute the opinion text to just a single judge. However, a substantial literature shows that bargaining over opinion content and panel effects influence U.S. Courts of Appeals decisions. Thus, there are within-panel dynamics where it is possible, if not likely, that multiple judges leave their fingerprints on the majority opinion.

Also, the authors assume that all parties bring all relevant arguments to the court’s attention in their briefs. I am not sure that this is necessarily the case, especially given there is reason for the parties to strategically select/push particular arguments. Perhaps the two parties combined will cover the full terrain in the end. But, given the evidence (and reason) to expect strategic attorneys, I have some doubt about the authors’ contention that the parties feel obligated to cover all the argument bases.

-- The authors give no consideration to the role of law clerks in drafting opinions. Is there consistency in polarization across a judge’s tenure? Some literature has used a lack of opinion text consistency over time as an indication that clerks may be more involved in drafting opinions. Of course, there may be other reasons why judges’ opinion text changes over time. But, as it stands now, I see no indication that the authors have considered how law clerks might undermine their research design and empirical results.

-- The paper would benefit from more substantive explanation of how the authors’ measurement approach translates text into an indicator of polarization. For instance, explain in less technical terms how the machine-learning approach yields a measure that meaningfully conveys judges’ propensity to use politically motivated language in their opinions.

-- Perhaps I missed it, but it’s not clear to me why the authors do not use concurring opinions in their analysis.

-- The paper would benefit (if space permits) from a bit more background theoretical discussion of why judges (as humans) engage in motivated reasoning.

Reviewer #2: This article offers a potentially interesting contribution, analyzing the extent to which federal courts of appeals have experienced partisan polarization over a very extended time period. The article is technically proficient, employing a suite of methods--particularly for the analysis of text--to analyze the extent to which language (or citations) predict the partisan affiliation of judges. However, while I found the paper interesting and was impressed by the clear work that went into this project, I left with a few concerns.

First, I found the manuscript lacking in its theoretical grounding around motivated reasoning. The authors introduce the concept, but they fail to establish a clear link between motivated reasoning and judicial behavior, to say nothing of the link between motivated reasoning, judicial behavior, and political partisanship. Indeed, by the latter half of the paper, much of the discussion of motivated reasoning is gone, and the discussion entirely centers on polarization, but the genesis of the switch between motivated reasoning to polarization is never made clear. Simply asserting that judges with different political orientations reason differently doesn’t connect well with the psychological literature around motivated reasoning. A more detailed exploration of motivated reasoning, and how it connects here--including consideration of things like cognitive biases--would substantially improve this. In all, I found the theoretical framing not to be a compelling link.

Second, and related, the lack of a cohesive theoretical framework compounds the challenges for the authors in measurement and in interpreting their empirical results. On the measurement side, capturing motivated reasoning as the extent to which party affiliation is predictable by language requires significantly more justification; shifting that to then be an indicator for political polarization requires yet more. On the empirical results, the lack of a theoretical framework and the underjustified measurement create a lot of uncertainty for interpretation. For instance, for divided panels, the authors find that polarization in texts and citations is lower, but that judges sitting on these divided panels are more likely to dissent. There’s no real story around why any of this might be the case. The authors note a general idea about heightened scrutiny on divided panels, but the explanation is not sufficiently developed from the earlier theoretical discussions. Moreover, why is the dissent separate at all? If the idea is that divided panels encourage scrutiny, and this precipitates each of these changes in polarization, shouldn’t the dissent be another component of scrutiny? The authors could improve this, again, by building a real theoretical framework from motivated reasoning, but at present that is absent.

Finally, the data here are impressive. However, the authors seem to really miss an opportunity to explore the over-time institutional and political transformations that occurred within the federal courts of appeals over this period. Indeed, there are major works in political science and legal studies on the changing partisan composition of the federal courts over this time period. Because the focus is on changing polarization, the absence of these discussions felt particularly pronounced. How do the observed changes match with the changing institutional parameters of the federal courts of appeals? Are there institutional predicates for the observed changes in polarization? Without engaging with these developments, the analysis holds almost entirely on an undertheorized story around individual judges polarizing.

6. PLOS authors have the option to publish the peer review history of their article (what does this mean?). If published, this will include your full peer review and any attached files.

Reviewer #1: No

Reviewer #2: No

---

## [Author Response · Author response to Decision Letter 1]

28 Oct 2024

Dear Dr. Gutmann and reviewers,

We appreciate your thoughtful comments and suggestions. In response, we have prepared a detailed point-by-point reply addressing each of their concerns and suggestions, which we have included in the attached response document. We believe these revisions have strengthened our manuscript, and we look forward to any additional feedback.

Thank you for the opportunity to improve our work.

Kind regards,

Wei and Daniel

---

## [Decision Letter · Decision Letter 1]

10 Dec 2024

PONE-D-24-34043R1Motivated Reasoning in the Field: Polarization of Prose, Precedent, and Policy in U.S. Circuit Courts, 1891-2013PLOS ONE

Dear Dr. Lu,

Thank you for submitting your manuscript to PLOS ONE. After careful consideration, we feel that it has merit but does not fully meet PLOS ONE’s publication criteria as it currently stands. Therefore, we invite you to submit a revised version of the manuscript that addresses the points raised during the review process.

We look forward to receiving your revised manuscript.

Kind regards,

Jerg Gutmann

Academic Editor

PLOS ONE

Journal Requirements:

**Additional Editor Comments:**

Unfortunately, reviewer #2 was at this time not available to review the revised version of your manuscript as well as your response to their suggestions. I have carefully read again the arguments brought forward by reviewer #2 and your reponse, and I considered these issues adequately addressed. Reviewer #1 is also overall satisfied with your efforts. However, they criticize your response to their previous second argument as not yet fully developed and convincing. In brief, the reviewer is wondering what makes the judges you study care about additional scrutiny. Maybe you could elaborate a bit more on this issue. Alternatively, you could cite any literature that makes similar assumptions about comparable judges' motivation. Or maybe it is possible to refer to some qualitative evidence supporting the plausibility of your assumptions.

Reviewers' comments:

Reviewer's Responses to Questions

**Comments to the Author**

1. If the authors have adequately addressed your comments raised in a previous round of review and you feel that this manuscript is now acceptable for publication, you may indicate that here to bypass the “Comments to the Author” section, enter your conflict of interest statement in the “Confidential to Editor” section, and submit your "Accept" recommendation.

Reviewer #1: (No Response)

2. Is the manuscript technically sound, and do the data support the conclusions?

Reviewer #1: Yes

3. Has the statistical analysis been performed appropriately and rigorously? 

Reviewer #1: Yes

4. Have the authors made all data underlying the findings in their manuscript fully available?

Reviewer #1: Yes

5. Is the manuscript presented in an intelligible fashion and written in standard English?

Reviewer #1: Yes

6. Review Comments to the Author

Reviewer #1: “Motivated Reasoning in the Field: Polarization of Prose, Precedent, and Policy in U.S. Circuit Courts, 1891-2013”

I appreciate the authors’ engagement with my original review. On balance, I am satisfied with their responses and revisions. I think they have meaningfully improved the paper and thus I recommend publication. However, in my view, there is one critique/response that would still benefit from more attention from the authors.

I think the authors’ argument related to election-year effects still lacks a clear theoretical mechanism. The central argument that judges want to appear less partisan or avoid attracting scrutiny makes some sense on its face, but I don’t see any discussion of why judges should care. That is, why do unelected judges with life tenure care about appearing less partisan or avoiding scrutiny? I think the conditional effect for judges facing promotion incentives makes some sense and is a partial answer. But, it still doesn’t fully address the question of why the average circuit judge, on balance, cares about avoiding scrutiny. And, furthermore, I am still skeptical that anyone cares all that much about scrutinizing many of these cases (i.e., perhaps this argument and empirical effect ought to be entirely conditional on certain salient cases). So, while I appreciate the authors’ engagement on this point, I still have some questions about this original critique.

7. PLOS authors have the option to publish the peer review history of their article (what does this mean?). If published, this will include your full peer review and any attached files.

Reviewer #1: No

---

## [Author Response · Author response to Decision Letter 2]

16 Jan 2025

Dear reviewer 1,

Thank you for your comment. We have addressed your concern in the paper and you can find the changes and explanations in the response letter.

Best,

Wei Lu and Daniel Chen

---

## [Editor Report · Decision Letter 2]

22 Jan 2025

Motivated Reasoning in the Field: Polarization of Prose, Precedent, and Policy in U.S. Circuit Courts, 1891-2013

PONE-D-24-34043R2

Dear Dr. Lu,

We’re pleased to inform you that your manuscript has been judged scientifically suitable for publication and will be formally accepted for publication once it meets all outstanding technical requirements.

Kind regards,

Jerg Gutmann

Academic Editor

PLOS ONE
---

## [Editor Report · Acceptance letter]

PONE-D-24-34043R2

PLOS ONE

Dear Dr. Lu,

I'm pleased to inform you that your manuscript has been deemed suitable for publication in PLOS ONE. Congratulations! Your manuscript is now being handed over to our production team.

Kind regards,

on behalf of

Prof. Dr. Jerg Gutmann

Academic Editor

PLOS ONE